# The Discovery of Actinospene, a New Polyene Macrolide with Broad Activity against Plant Fungal Pathogens and Pathogenic Yeasts

**DOI:** 10.3390/molecules26227020

**Published:** 2021-11-20

**Authors:** Ying Tang, Cuiyang Zhang, Tianqi Cui, Ping Lei, Zhaohui Guo, Hailong Wang, Qingshu Liu

**Affiliations:** 1Hunan Institute of Microbiology, Changsha 410009, China; tying_1986@126.com (Y.T.); zhang_cuiyang@163.com (C.Z.); leiping2021@126.com (P.L.); 2State Key Laboratory of Microbial Technology, Institute of Microbial Technology, Helmholtz International Lab for Anti-infectives, Shandong University–Helmholtz Institute of Biotechnology, Shandong University, Qingdao 266237, China; ctqlsjp@163.com

**Keywords:** polyene macrolide, actinospene, genome mining, *Actinokineospora spheciospongiae*, antifungal activity, cryptic gene cluster

## Abstract

Phytopathogenic fungi infect crops, presenting a worldwide threat to agriculture. Polyene macrolides are one of the most effective antifungal agents applied in human therapy and crop protection. In this study, we found a cryptic polyene biosynthetic gene cluster in *Actinokineospora spheciospongiae* by genome mining. Then, this gene cluster was activated via varying fermentation conditions, leading to the discovery of new polyene actinospene (**1**), which was subsequently isolated and its structure determined through spectroscopic techniques including UV, HR-MS, and NMR. The absolute configuration was confirmed by comparing the calculated and experimental electronic circular dichroism (ECD) spectra. Unlike known polyene macrolides, actinospene (**1**) demonstrated more versatile post-assembling decorations including two epoxide groups and an unusual isobutenyl side chain. In bioassays, actinospene (**1**) showed a broad spectrum of antifungal activity against several plant fungal pathogens as well as pathogenic yeasts with minimum inhibitory concentrations ranging between 2 and 10 μg/mL.

## 1. Introduction

Fungal infections on crops are the major cause for the economic losses in global agriculture [1]. Phytopathogenic fungi could result in root rot or blackening, wilting, and even plant death, thus decreasing crop yield and quality. The effective control of plant fungal pathogens currently mainly relies on synthetic fungicides. However, excessive usage of fungicide could lead to the emergency of resistance and concerns over the residual effects on the environment [2]. Furthermore, due to the poor soil accessibility, chemical fungicides showed incompetent performance in the control for soil-borne pathogens, as they could survive in soil through the formation of oospores, chlamydospores, and sclerotia [3].

Biological control is a promising and sustainable approach for managing plant fungal diseases. Microbial biocontrol agents for plant diseases are usually fungal or bacterial strains, which could prevent the infection of the host plant by the pathogen via various mechanisms including antibiosis, mycoparasitism, induced resistance, and growth enhancement. Among them, the production of an inhibitory metabolite or antibiotic is one of the most important mechanisms, as demonstrated by the successful application of many strains in genus *Streptomyces* [4,5], *Bacillus* [6], and *Pseudomonas* [7], where the production of polyene macrolides, lipopeptides, and toxins—phenazine-1-carboxylate, for example—are indispensable for their biocontrol activity, respectively.

The dramatic expansion of genomic information has uncovered a large number of biosynthetic gene clusters (BGCs). Many of them remain silent under laboratory conditions due to the absence of environmental triggers or stimuli. Benefiting from the deep understanding toward the chemical logic and enzymatic machinery for the biological assembly of natural products and the profound improvement in the bioinformatics, a range of genome mining approaches have been devised and consequently shown to be an efficient strategy for the rational discovery of a targeted group of compound [8].

Polyene macrolides are one of the most effective group of antifungal agents [9]. Historically, they have been the first antibiotics used in antifungal therapy, as exemplified by nystatin and amphotericin discovered in the 1950s [10,11]. The rare occurrence of resistance due to a unique ergosterol-binding mechanism, and potential activity against enveloped viruses, parasites, pathogenic prion protein, and carcinoma cells, emphasizes the importance of polyene macrolide as pharmaceuticals, although they are known for their debilitating toxicity and poor distribution [12]. Apart from their remarkable application as a therapeutic agent on humans, polyene macrolides also exhibit great potential in the integrated management of plant fungal pathogens [4,5,13,14].

The core structure of polyene macrolides is synthesized by type I modular polyketide synthases. A spontaneous formed internal hemiketal ring and a carboxyl group converted from the methyl branch on this ring are the common structural features for almost all polyenes. Other characteristic late modifications such as hydroxylation, epoxidation, and glycosylation will selectively happen [15]. Currently, most of the polyenes isolated from nature and widely used in clinics and industries are glycosylated, which is apparently due to their considerably better activity/toxicity ratio compared to the non-glycosylated polyene macrolides [9]. Two deoxyaminosugars, mycosamine and perosamine, are exclusively found in the glycosylated polyene macrolides [16]; both require the GDP-mannose-4,6-dehydratase during their biosynthesis. Accordingly, the polyene macrolide BGCs identified so far all contain a gene encoding the GDP-mannose-4,6-dehydratase. Therefore, we reason that GDP-mannose-4,6-dehydratase could be used as a beacon for mining new polyene macrolides.

Here, we report the genome mining-guided discovery, activation, isolation, structural elucidation, and biological activity of actinospene (Figure 1 and Appendix A), which is a unique polyene macrolide containing two epoxy groups and an isobutenyl side chain from *Actinokineospora spheciospongiae*, which was previously isolated from marine sponge tissue [17,18,19].

## 2. Results and Discussion

### 2.1. Genome Mining and Activation of the Actinospene Biosynthetic Gene Cluster

We chose the sequence of the GDP-mannose-4,6-dehydratase encoded by *NysD*III in the biosynthetic pathway for nystatin from *Streptomyces noursei* ATCC 11455 to mining the potential producer of glycosylated polyene against the public available genome via the assistance of HMMER web server [20]. This query sequence picked out plenty of actinomyces species, which all contained a GDP-mannose-4,6-dehydratase highly similar to NysDIII. Most of them belong to the genus *Streptomyces*, rare actinomyces such as genus *Pseudonocardia*, *Amycolatopsis* occupy a small portion. We eliminated all the *Streptomyces* strains, as most of the known polyene macrolides are from this genus [15]. Several rare actinomycetes including *A. spheciospongia**e* were cultivated under different fermentation conditions for screening the corresponding product. Out of more than 20 tested media, only the extract from *A. spheciospongiae* showed the anticipated antifungal activity when cultivated in SFM medium. Accordingly, a unique peak at 310 nm with the characteristic UV-Vis spectrum of three peaks for polyene macrolide could also be found exclusively in this extract when comparing the metabolic profile of *A. spheciospongiae* fermented under other media (Figure 2 and Appendix A).

### 2.2. Isolation and Structure Elucidation

To purify the putative new antifungal compound, a large-scale fermentation was performed and extracted. Guided by the bioactivity and the distinctive peak, compound **1** was purified using silica gel column, Sephadex-LH20 column chromatography, and semi-preparative HPLC in sequence, yielding 10 mg of **1** with good purity.

The molecular formula of actinospene (**1**) was determined to be C_38_H_55_NO_15_, as indicated by the protonated molecular ion at *m/z* 766.3655[M + H]^+^ (calcd. 766.3644) from HRMS (Appendix A, see Appendix A). The planar structure was elucidated using data from 1D and 2D NMR techniques (Table 1, Appendix A).

The ^13^C NMR spectrum exhibits 38 carbon signals, which could be assigned to two carbonyls (C-1, C-29), five methyls (C-6′, C-28, C-30, C-31, C-32), three methylenes (C-6, C-8, C-14), twenty-five methines, and three quaternary carbons (C-9, C-20, C-26) according to HSQC and DEPT experiments. Further interpretation of the 2D NMR data revealed the presence three vinyl methyls, eight olefinic protons, fourteen oxymethine, and an aminosugar moiety. The ^1^H-^1^H COSY spectrum displayed all expected connectivities within four structural blocks C-2–C-8, C-10–C-19, C-21–C-25, C-27–C-28, and the perosamine moiety. These blocks were assembled based upon the HMBC experiment, which showed long-rang heteronuclear connectivities. The location of the lactone moiety was determined based on the H-2/C-1 and H-25/C-1 correlations. The placement of the glycosidic bond at C-15 resulted from H-1′/C-15 connectivity. The blocks C-10–C-19 and C-21–C-25 were connected via C-20 according to the H-19/C-20 and H-21/C-19 correlations. The H-25/C-26 and H-28/C-26 connectivities confirmed the attachment of block C-27–C-28 to C-21–C-25, while the H-8/C-9 and H-10/C-8 correlation revealed the linkage between blocks C-2–C-8 and C-10–C-19 via C-9 (Figure 3a).

The geometry of the three double bonds of the tetraene chromophore has been assigned as 16*E*, 18*E*, and 22*E*-configuraiton based on the large vicinal coupling constants between H-16 and H-17 (*J* = 14.9 Hz), H-18 and H-19 (*J* = 15.0 Hz), H-22 and H-23 (*J* = 13.7 Hz) [21]. The vicinal coupling constant *J*20,21 could not be detected due to the absence of proton on C-20. However, NOESY correlation between Me-30 and H-22 supported the geometry of the double bond between C-20 and C-21 to be *E* (Figure 3b). Meanwhile, the configuration of the double between C-26 and C-27 was supposed to be *cis* based on the chemical shift of C-28 and C-32 [22]. The relative configuration of the hemiketal ring (C-9~C-13) was revealed as a chair conformation with H-10, H-11, H-12, and H-13 being in the axial positions according to NOESY signals observed between H-11 and H-13, H-10 and H-12 (Figure 4). The sequential COSY correlations from H-1′ to Me-6′ and the HMBC correlations of H_3_-6′ with C-4′ and C-5′ implied the existence of the perosamine sugar moiety. The NOESY interactions between H-1′ and both H-3′/H-5′ were indicative for the equatorial position of the 3′-OH and 6′-Me. The missing correlation between H-5′ and H-4′ suggested 4′-NH_2_ to be in an equatorial position, too. Meanwhile, the NOE correlations of H-1′ with H_2_-14/H-17/H-13, and the absence of the NOESY signal between H-13 and H-15, indicated the sugar to be connected to C-15 via β-type glycosidic linkage (Figure 4).

### 2.3. Proposed Model for Biosynthetic Pathway of Actinospene

*A. spheciospongiae* contains one homologous protein to the NysDIII with an E-value of 9.6e-185, and it was designated as ActnJ. According to sequence analysis via antiSMASH 6.0 [23], only one gene cluster with the module organization highly collinear to the chemical structure of actinospene, and containing all the accessory genes for the post assembling modifications, could be found in its genome. However, unlike other formerly reported polyene biosynthetic pathways, this gene cluster is not adjacent to the gene *ActnJ*, which encodes the GDP-mannose-4,6-dehydratase. The gene cluster region has a size of approximately 129 kb and contains six genes encoding typical multifunctional type I PKSs, four genes encoding cytochrome P450 monooxygenases, two genes responsible for perosamine biosynthesis and attachment, three genes involved in transcriptional regulation, and several genes encoding ABC transporters (Appendix A).

The first module in the ActnS0 protein was deduced as a loading module, because it contains a ketosynthase (KS^S^) domain similar to those found in the loading modules of the pimaricin PKS PimS0 [14], nystatin PKS NysA [24], amphotericin PKS AmphA and tetramycin PKS TetrA [25,26], in which the conserved active-site cysteine residue is replaced with a serine residue (Appendix A). The arrangement of other modules was settled by the substrate specificity of the AT domains and the decoration domains.

According to the multiple sequence alignments, all AT domains possess a GHSXG motif at the active site. The AT domains ActnS2AT3 and ActnS3AT7 are predicted to be specific for methylmalonyl-CoA due to the presence of a YASH motif. The YGSH motif containing ActnS0AT1 is assigned to be methylmalonyl-CoA specific too based on antiSMASH analysis. The remaining AT domains all have the HAFH motif; thus, they are malonyl-CoA specific (Appendix A) [27].

The 26-membered macrocyclic lactone of actinospene requires 13 rounds of extension; however, only 11 modules reasonable for the biosynthesis of **1** could be found in the gene cluster. Therefore, we reckon that the one module encoding ActnS1 might catalyze three rounds of elongation and β-keto processing. The biosynthesis of several modular PKs such as the stigmatellin and aureothin has been reported to be able to use individual module iteratively for twice or more times [28,29].

The observed hydroxylation pattern at C-25 (atom numbering according to NMR assignment, Table 1) demanded the DH domain ActnS0DH1 to be inactive or skipped during the assembly of the PKS core structure. Full-length sequence analysis of the DHs demonstrated that the ActnS0DH1 harbored the “conserved” amino acid sequence and thus should be functional (Appendix A). Similar active DH domains that are retained in the cluster but serve no function could also be found in the NysA for nystatin [24] and the TrmB for termidomycin [30].

The mature polyketide chain is presumed to be cleaved from the PKS complex by the thioesterase domain in ActnS4; then, it is cyclized and subjected to further decorations. Based on the phylogenetic analysis of the P450 monooxygenase (Appendix A), ActnG showed a close evolutionary relationship to TetrG, which has been demonstrated to be responsible for generating the carboxylic acid group of tetramycin [25]. Downstream of the TetrG, the ferredoxin encoding gene TetrF is located. A similar gene arrangement exists also in biosynthetic gene clusters for many other polyene macrolides, including the actinospene [14]. Therefore, ActnG and ActnF are probably involved in the oxidation of the methyl group at C-12 of actinospene. ActnD2 and ActnD3 fall into the same branch with the NysL and AmphL, which have been shown to catalyze the hydroxylation on the polyol region; thus, the hydroxyl group on C-10 should be introduced by either ActnD2 or ActnD3. The remaining ActnD1 was proposed to be engaged in the oxidation of the two double bonds between C-4 and C-5, C-2 and C-3 based on the phylogenetic analysis (Appendix A).

Alkyl branches at positions corresponding to former acetyl carboxyl groups (C1) have been observed in several polyketides [31]. They are usually introduced by 3-hydroxy-3-methylglutaryl-CoA (HMG) synthase (HCS) and enoyl-CoA hydratase (ECH, or crotonase) homologues [32]. Actinospene (**1**) contains an isobutenyl group on C-25. Yet, a HCS homologue could not be found in the genome; instead, two genes encoding acyl-CoA ligase and dehydrogenase were detected in the actinospene gene cluster. Therefore, we speculated that they should be responsible for the attachment of this side chain. Finally, a perosamine would be synthesized and attached to the macrocyclic aglycone on C-15 by ActnC, ActnK, and ActnJ (Figure 5).

### 2.4. In Silico Prediction of Stereochemistry of Actinospene Core Structure

The ketoreductase (KR) domains are responsible for setting both the β-hydroxyl group and α-substituent stereochemistries in polyketides. According to their stereo-selectivity, KRs could be classified into six types, and each possesses specific sequence motifs [33]. Therefore, the configurations of hydroxy- and methyl-bearing chiral centers could be deduced from multiple sequence alignment. This method has been used for the stereo-chemical assignments of a number of structurally complex polyketides, such as ajudazols [34], niphimycins [35], and thailandamide A [36], many of which were later verified by chemical methods.

Due to the absence of a conserved LxD (LDD) motif in the loop region, and the presence of the conserved tryptophan and histidine in the catalytic region, ActnS3_KR7 was unambiguously assigned as an A2 type that can epimerize the α-methyl group from “R” to an “S” configuration (Appendix A). The conserved tryptophan in the catalytic region was replaced by leucine or isoleucine in ActnS3_KR6 and ActnS3_KR8 (Appendix A). Similar replacement also exists in the PimS2KR8, NysIKR12, which has been prove to be responsible for an “S” hydroxyl group. Thus, we proposed these two KRs to be A1-type (Appendix A). The rest of the KRs were designated as the B1 type in view of the LDD motif and the missing proline in the active site (Appendix A). Although the ActnS2KR3 owns an IDD motif instead of the LDD, the large J coupling constant observed between H-22 and H-23 supported it to be a B1 type.

Then, the absolute configuration of **1** was confirmed by comparing the experimental electronic circular dichroism (ECD) spectrum of **1** with the theoretically calculated spectra of both enantiomers (Figure 6). Thus, the 2S, 3R, 4S, 5R, 7S, 10S, 11R, 12R, 13S, 15S, 24S, and 25S were assigned to **1**. A similar approach has been applied for stereo configuration determination of several recently published compounds [37,38,39].

### 2.5. Bioactivity Analysis of the Actinospene

An initial agar diffusion susceptibility test using the crude extract from *A. spheciospongiae* cultivated in SFM medium demonstrated good activity against a variety of pathogenic fungi (Appendix A). To fully investigate the inhibition activity of actinospene against fungal cells, three yeast strains and six filamentous plant pathogenic fungi were chosen as indicators to evaluate the minimal inhibitory concentration (MIC). As shown in Table 2, actinospene showed good activity against most of the tested yeast and filamentous fungi with MIC values ranging between 2 and 10 μg/mL, which is comparable to the values of pimaricin, a known antifungal polyene antibiotic widely used in food, beverage, and crop protection [14]. As for *Candida albicans*, *Fusarium graminearum,* and *Colletotrichum capsici*, actinospene displayed around four times higher MIC values when compared to pimaricin. In agreement with other polyene macrolides, actinospene showed no antimicrobial effect at concentration up to 64 µg/mL against the common indicator Gram-positive and Gram-negative strains.

The majority of the known polyene antibiotics are from the genus *Streptomyces*. Although bioinformatic analysis demonstrated that polyene biosynthetic gene clusters also existed in a great range of rare Actinomyces including *Actinokineospora*, *Amycolatopsis*, *Saccharopolyspora*, *Sinosporangium*, and *Salinispora*, the corresponding products have seldom been identified. In terms of the MIC values tested so far and the elucidated structure, it is evident that actinospene is eligible to be successfully exploited as agrochemicals and pharmaceuticals.

Based on former investigation, the marine sponge-derived *A. spheciospongiae* was recognized as a potential producer of fascinating chemical scaffolds [18,19]. Previously reported natural products from this strain included the antiparasitic actinosporins A−B, actinosporins C−D with antioxidant activity, antibacterial peptide actinokineosin, and fridamycins H-I [40,41]. The identification of actinospene (**1**) expanded our understanding toward its metabolic capacity. Future verification of the proposed biosynthetic pathway might reveal new mechanisms for the biosynthesis and regulation of polyene macrolides.

## 3. Materials and Methods

### 3.1. General Experimental Procedures

All reagents used in this study were HPLC grade or analytical grade solvent. Infrared spectra were acquired using a Thermo Nicolet iS10 IR spectrometer. CD spectra (ca. 0.5 mg/mL in methanol) were tested by a MOS-500 CD spectropolarimeter (Bio-Logic, France) in a 0.1 mm cuvette. Optical rotations were measured on an AUTOPOL III instrument (Rudolph Research Analytical, Hackettstown, USA). Nuclear magnetic resonance (NMR) was measured on a 700 MHz Avance III (Ascend) spectrometer by Bruker BioSpin GmbH, equipped each with a cryoplatform, at 25 °C. Spectra were recorded in 500 μL DMSO-d_6_. Solvent signals were used as an internal standard (DMSO-d_6_: δ_H_ 2.50, δ_C_ 39.5 ppm). High-resolution mass spectrometry was carried out on Agilent 6545 Quadrupole Time of Flight (Q-TOF) high-resolution mass spectrometer equipped with a reverse-phase C18 column (Agilent, Eclipse Plus, 50 × 2.1 mm, 1.8 μm), running in positive ionization mode with a resolution of 30,000. The flow rate was set at 0.3 mL/min with a mobile phase of H_2_O/ACN each containing 0.1% of formic acid. The ACN percentage gradually increased from 5% to 95% in 12 min. The injection volume was 2 μL. Selected ions were fragmented using a collision-induced dissociation energy of 40 eV.

### 3.2. Fermentation, Extraction, and Isolation

The spore of *A. spheciospongiae* was first inoculated into GYM and shaken at 150 rpm for 7 days at 28 °C. Then, this seed culture was transferred onto SFM medium and incubated at 28 °C for 15 days. The extract was subsequently filtered using 8 layers of gauze and evaporated in a rotary evaporator under vacuum (IKA, Königswinter, Germany). Initial separation was performed on a normal-gel silica gel column (100–200 mesh, Qingdao Haiyang Chemical Group Co., Qingdao, China) using a step elution with combinations of hexane, EtOAc, and MeOH (hexane/EtOAc, 1:1; 100% EtOAc; EtOAc/MeOH, 1:1; and 100% methanol). The fraction with antifungal activity was further purified on a Sephadex LH-20 (GE Pharmacia, USA) column (100 × 2.5 cm; flow rate 0.025 L/h over 36 h) using methanol as the mobile phase. Fractions containing significant amounts of **1** were collected, dried, redissolved, and subjected to a semi-preparative reverse-phase HPLC system (Agilent ZORBAXSB-C18, 9.4 × 250 mm, 5 μm, DAD at 310 nm) running with a MeOH-H_2_O gradient solvent system (40% MeOH/H_2_O for 4 min, then 40–95% MeOH/H_2_O for 30 min, 3 mL/min flow rate). The fraction eluting from 27.4 min to 28.4 min was collected yielding 10 mg of 1 as a light grayish white amorphous solid.

Actinospene (**1**): light grayish white amorphous solid. [α]^25^_D_-18 (c 0.75, MeOH); UV (MeOH) *λ*_max_ 310 nm; IR (ATR) *v*_max_ 3430, 1740, 1630, 1390, 1101, 990, 580 cm^−1^; ^1^H and ^13^C NMR, Table 1; HRESI/MS: *m/z* 766.3655 [M + H]^+^ (calculated for C_38_H_55_NO_15_, 766.3644).

### 3.3. ECD Calculations

Systematic conformational analysis was performed by using the MMFF94 force field calculator. All the conformers were optimized with the software package Gaussian 09 (Gaussian Inc., Wallingford, NY, USA) first using the semi-empirical method PM6 and then the quantum mechanical DFT method B3LYP/6-31G/methanol, which was also used to calculate its ECD transitions (TDDFT). The line spectrum conformations were built by applying a Gaussian line broadening of 0.4 eV. The overall theoretical ECD spectra were obtained according to Boltzmann weighting of each conformers [42].

### 3.4. Antifungal Activity Assays

Antifungal activities of the crude extracts, fractions, and pure compounds were detected using the agar diffusion susceptibility test [13,43]. The indicator fungi are *Candida albicans*, *Cryptococcus neoformans*, *Saccharomyces cerevisiae*, *Fusariums oxysporum*, *Alternaria alternate*, *Fusarium graminearum*, *Sclerotium rolfsi*, *Phytophthora capsici*, and *Colletotrichum capsici*. Briefly, an aliquot of 90 μL extracts or fractions dissolved in DMSO was loaded onto the wells on Potato Dextrose Agar (PDA) previously coated with the indicate strains. The zone of growth inhibition was measured after incubation at 28 °C for 2–4 days for mold and 35 °C overnight for yeasts.

The minimal inhibitory concentrations (MICs) of actinospene were determined using the broth dilution method in 96-well microplates according to the European Committee on Antimicrobial Susceptibility Testing (EUCAST) definitive document EDef 7.2, 9.3, and 11.0 [44,45,46]. The inoculum suspensions were diluted with RPMI 1640 broth for plant pathogenic fungi and Potato Dextrose Broth (PDB) for yeast to achieve a final inoculum density of 0.4–5 × 10^4^ CFU/mL and 5 × 10^5^ CFU/mL, respectively. The final volume in each well was 200 μL. Compounds dissolved in DMSO (Sangon Biotech, Shanghai, China) were prepared using serial dilutions to obtain concentrations ranging from 256 to 0.125 μg/mL. For mold, the pimaricin was diluted using two-fold serial diltions from 88 μg/mL, while the amphotericin was from 10 μg/mL. The concentrations of actinospene against *Fusariums oxysporum* and *Colletotrichum capsici* were set at 5, 10, 20, 30, 40, 50, 60, and 70 μg/mL. Blank medium was used as the sterility control. DMSO alone at the same concentration was carried out as a negative control. The pimaricin and amphotericin B (Sangon Biotech, Shanghai, China) were chosen as a positive control. All experiments were performed in triplicate. The MICs were recorded as the lowest concentration of the compounds that caused complete growth inhibition.

### 3.5. Microorganisms and Culture Conditions

The *A. spheciospongiae* DSM45935 was obtained from German Collection of Microorganisms and Cell Cultures GmbH (DSMZ, Braunschweig, Germany). *Staphylococcus aureus* GDMCC 1.2442, *Escherichia coli* GDMCC 1.1478, *Candida albicans* GDMCC 2.194, and *Saccharomyces cerevisiae* GDMCC 2.73 were purchased from Guangdong Microbial Culture Collection Center (GDMCC, Guangzhou, China). *Cryptococcus neoformans* H99 was kindly provided by Dr. Yi Zou, Southwest University, China. The plant fungal pathogens *Fusariums oxysporum, Alternaria alternate, Fusarium graminearum, Sclerotium rolfsi, Phytophthora capsici,* and *Colletotrichum capsici* are generously offered by Dr. Wu Chen, Hunan Agriculture University, China. *S. aureus* and *E. coli* are grown routinely on LB agar plates or in LB liquid medium at 37 °C. All fungus strains were cultivated and maintained on Potato Dextrose Agar (PDA) at 28 °C.

### 3.6. Bioinformatics

The HMMER web server was employed to screen strains containing the polyene macrolide biosynthetic gene cluster using NysDIII as the query sequence [20]. Then, the whole genome sequence (accession number: GCA_000564855.1) of the spotted strains was then detected for polyene biosynthetic gene clusters using the online software antiSMASH 6.0 [23]. Multiple sequence alignment of the ketoacyl synthase (KS), acyl transferase (AT), β-ketoacyl reductase (KR), and β-hydroxyacyl dehydratase (DH) domains were conducted as previously described employing Clustal Omega [47]. The phylogenetic tree of the P450 monooxygenases was reconstructed using the MEGA7 program [48] by the neighbor-joining method [49]. Evolutionary distances were calculated using Kimura’s two-parameter model, and bootstrap values were calculated based on 1000 replications.

## 4. Patents

We have applied a patent concerning the structure and application of actinospene in the China National Intellectual Property Administration. The application number is 202110968567.8.

## Figures and Tables

**Figure 1 molecules-26-07020-f001:**
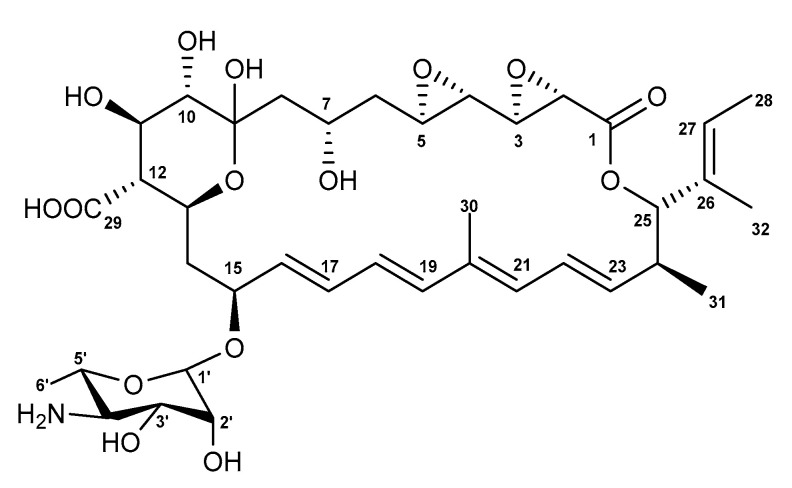
Chemical structure of actinospene (**1**).

**Figure 2 molecules-26-07020-f002:**
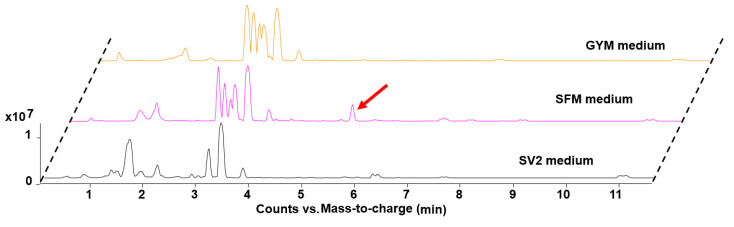
Comparison of metabolite profiles of the *A. spheciospongiae* fermented in SFM, GYM, and SV2 medium, respectively.

**Figure 3 molecules-26-07020-f003:**
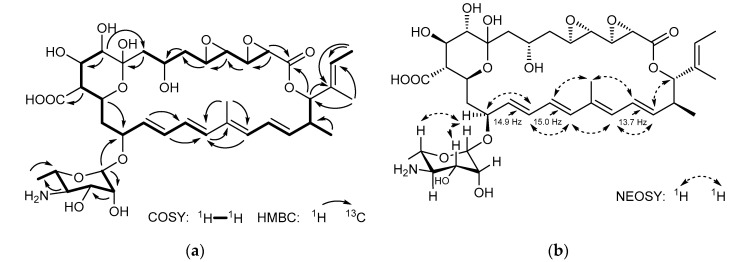
Planar structure of actinospene (**1**) showing 2D NMR correlations. (**a**) Observed COSY and HMBC correlations; (**b**) NOESY correlations, and the diagnostic ^1^H-^1^H coupling constants.

**Figure 4 molecules-26-07020-f004:**
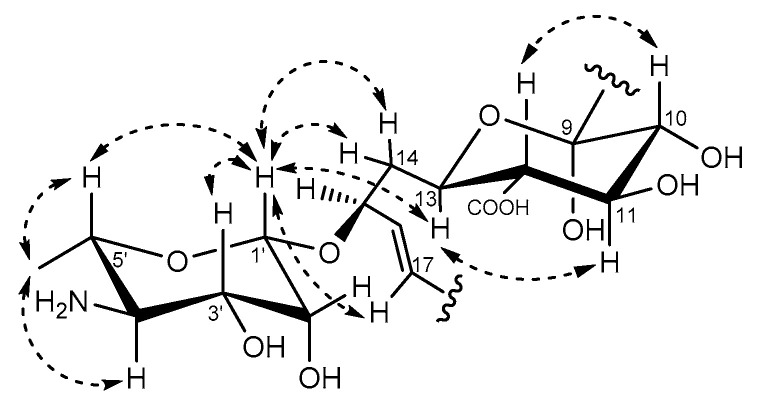
Relative configuration assignment of the hemiketal ring and the perosamine in actinospene (**1**) based on NOESY correlations.

**Figure 5 molecules-26-07020-f005:**
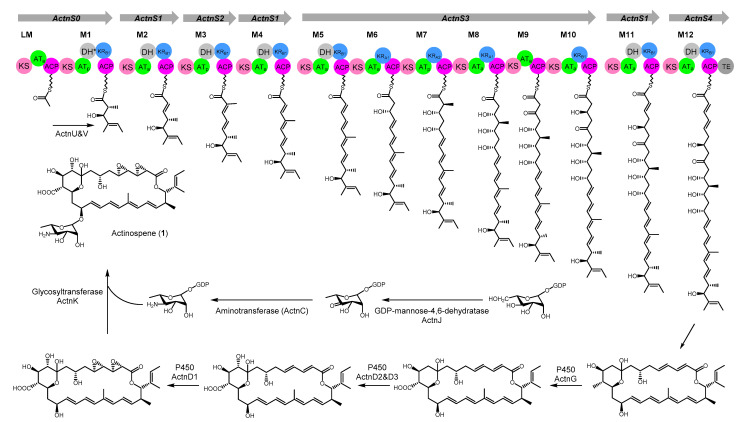
Proposed biosynthetic pathway for actinospene. Genes are shown as gray arrows, and modules of the biosynthetic machinery each containing a set of catalytic domains illustrated as circles (KS, ketoacyl synthase; AT, acyl transferase; ACP, acyl carrier protein; KR, β-ketoacyl reductase; DH, β-hydroxyacyl dehydratase; TE, thioesterase). AT and KR domains are shown with predicted stereochemistry based on sequence alignment (ATa, malonyl-CoA; ATp, methylmalonyl-CoA; KR_A1_, A1-type ketoreductase; KR_A2_, A2-type ketoreductase; KR_B1_, B1-Type ketoreductase). * indicates an inactive DH domain.

**Figure 6 molecules-26-07020-f006:**
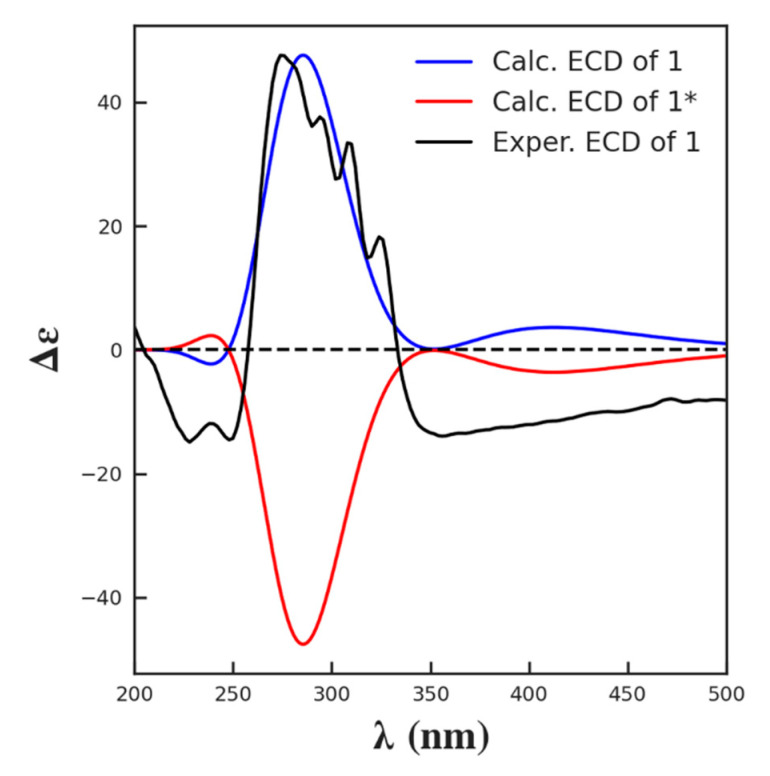
Experimental methanol ECD spectrum of **1** (black line) and theoretical ECD spectra of **1** (blue line) and its enantiomer **1***(red line).

**Table 1 molecules-26-07020-t001:** NMR spectroscopic data for actinospene (1) in DMSO-d_6_ (δ_H_, 600 MHz; δ_C_, 150 MHz).

Carcon no.	Actinospene (1)			
	δ_C_	δ_H_ (*J* in Hz)	COSY	HMBC
1	166.6, C			
2	50.5, CH	3.64, d^a^	3	1, 3, 4
3	56.7, CH	2.45, dd^a^	2, 4	4
4	54.4, CH	2.33, d (7.6)	3, 5	5
5	53.3, CH	2.59, m	4, 6	3, 6
6	39.9, CH_2_	1.95, m^a^	5, 7	5, 7
		0.94, m	5, 7	
7	66.0, CH	3.98, m	6, 8	
8	42.7, CH_2_	1.84, m^a^	7	7, 9
		1.39, d (13.9)	7	7, 9
9	98.5, C			
10	75.7, CH	2.82, d (8.8)	11	8, 11
11	69.8, CH	3.73, dd (8.8, 10.8)	10, 12	10, 12, 29
12	56.9, CH	1.95, m^a^	11, 13	11, 13, 14, 29
13	64.8, CH	4.20, m	12, 14	15
14	37.0, CH_2_	2.19, d (13.1)	13, 15	
		1.49, m	13, 15	
15	74.6, CH	4.37, m	14, 16	
16	136.4, CH	5.87, dd (8.6, 14.9)	15, 17	18
17	128.6, CH	6.07, dd (11.0, 14.9)	16, 18	15, 18, 19
18	129.3, CH	6.47, dd (11.0, 15.0)	17, 19	16, 17, 19, 20
19	135.6, CH	6.23, d (15.0)	18	17, 18, 20, 21, 30
20	134.6, C			
21	129.9, CH	6.00, d (12.0)	22	19, 22, 23, 30
22	128.8, CH	6.33, dd (12.0, 13.7)	21, 23	20, 21, 24
23	136.3, CH	5.33, dd (9.5, 13.7)	22, 24	21, 24, 31
24	39.5, CH	2.54, m^a^	23, 25, 31	
25	82.6, CH	4.85, d (10.4)	24	1, 23, 24, 26, 27, 31, 32
26	131.5, C			
27	126.3, CH	5.57, q (6.5)	28	25, 28, 32
28	13.0, CH_3_	1.59, d (6.5)	27	26, 27
29	174.6, C			
30	12.7, CH_3_	1.84, s		19, 20, 21
31	16.3, CH_3_	0.81, d (6.2)	24	23, 24, 25
32	10.6, CH_3_	1.55, s		25, 26, 27
1’	96.5, CH	4.41, s	2’	15, 2’
2’	69.8, CH	3.64, m^a^	1’, 3’	3’, 4’
3’	72.9, CH	3.22, d (8.9)	2’, 4’	
4’	54.6, CH	2.46, m^a^	3’, 5’	3’, 5’, 6’
5’	72.0, CH	3.08, m	4’, 6’	
6’	18.3, CH_3_	1.15, d (5.5)	5’	4’, 5’

Abbreviation: *s* = singlet, *d* = doublet, *dd* = doublet of doublets, *q* = quartet, *m* =multiplet, ^a^ overlapped.

**Table 2 molecules-26-07020-t002:** MICs of Actinospene (1) against pathogenic fungi and bacteria.

Species	MIC (μg/mL)		
	Actinospene	Pimaricin	Amphotericin B
*Saccharomyces cerevisiae*	2	2	0.25
*Candida albicans*	8	2	0.25
*Cryptococcus neoformans*	2	2	0.25
*Fusariums oxysporum*	10	11	5
*Sclerotium rolfsi*	2	1.38	0.02
*Fusarium graminearum*	8	1.38	0.63
*Colletotrichum capsici*	50	11	2.5
*Alternaria alternate*	8	5.5	0.16
*Phytophthora capsici*	4	5.5	0.16
*Staphylococcus aureus*	>64	NT	NT
*Escherichia coli*	>64	NT	NT

MIC values are shown in μg/mL. NT, not tested.

## Data Availability

The data presented in this study are available in article and Appendix A.

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
