# Peer review of "The Discovery of Actinospene, a New Polyene Macrolide with Broad Activity against Plant Fungal Pathogens and Pathogenic Yeasts"

_molecules, 2021, doi:10.3390/molecules26227020_

Round 1

Reviewer 1 Report

I have reviewed the manuscript entitled “The discovery of actinospene, a new polyene macrolide with 2 broad activity against plant fungal pathogens and pathogenic 3 yeasts.” By Tang et al. The manuscript is well designed, well written and organized. The results are novel.

During the reviewing process some minor comments were raised. 

3.4. Antifungal Activity Assays.

The authors incubate the plates at 28 °C for fungi and yeasts while yeast must be incubated at 35 °C not at 28 °C. The term fungi must be replaced by mold.

The reference Bonev et al., 2008 (40) is for antibacterial activity and not for antifungal activity assay. It is better to use.

EUCAST references for MIC is old (2012, 2014). Please update the new version of EUCAST.

In MIC, the authors dissolved the compounds in DMSO to obtain initial concentration μg/mL and then dilute the compound by 2-fold serial dilutions to obtain concentrations ranging from 256 to 0.5 μg/mL i.e. the results will be one of the following concentrations; 256, 128, 64,32,16,8,4,2,1,0.5 μg/mL while in the results of MIC (Table 2) I noticed 50, 11, 5, 5.5, 1.38, 0.25 etc. Please explain.

Reviewer 2 Report

Dear Authors,

The manuscript reported a new natural product was induced by using a different culture media. The structure was determined by 2D NMR, DFT calculation and bioinformatic analyses. Biosynthetic pathway was proposed based on genome sequence, and antimicrobial properties were evaluated. 

1) The large vicinal coupling constants values indicated trans configuration of olefin should be cited. There are a lot of such reference from MDPI journals. The 13C NMR of methyl carbon at C-30 and C-32 can also be used to deduce the geometry of an olefin, example olefin methyl carbon at >20 ppm might suggested a trans geometry.

2) Configuration of a double bond at C-26/C-27 should be indicated in text.

3) 10 mg is quite a lot, hydrolysis and comparison of sugar unit with standard could determine its absolute configuration. Mosher's method might be considered for secondary alcohols.

4) The NCBI accession number for the genome sequence of studied strain should be provided.

5) Proposed biosynthesis pathway could be experimental validate with gene knockout or heterologous expression. 

6) The tool for DFT calculation should be cited.
